# Materials and Technique: The First Look at Saturnino Gatti

Letizia Bonizzoni [1] , Simone Caglio [2],*, Anna Galli [2] , Luca Lanteri [3] and Claudia Pelosi [3]

1. Department of Physics Aldo Pontremoli, University of Milano, Via Giovanni Celoria, 16, 20133 Milano, Italy; letizia.bonizzoni@unimi.it
2. Department of Material Science, University of Milano Bicocca, Via Cozzi 55, 20125 Milano, Italy; anna.galli@unimib.it
3. Department DEIM, University of Tuscia, Largo dell'Università, 01100 Viterbo, Italy; llanteri@unitus.it (L.L.); pelosi@unitus.it (C.P.)
* Correspondence: simone.caglio@unimib.it

**Abstract:** As part of the study project of the pictorial cycle, attributed to Saturnino Gatti, in the church of San Panfilo at Villagrande di Tornimparte (AQ), image analyses were performed in order to document the general conservation conditions of the surfaces, and to map the different painting materials to be subsequently examined using spectroscopic techniques. To acquire the images, radiation sources, ranging from ultraviolet to near infrared, were used; analyses of ultraviolet fluorescence (UVF), infrared reflectography (IRR), infrared false colors (IRFC), and optical microscopy in visible light (OM) were carried out on all the panels of the mural painting of the apsidal conch. The Hypercolorimetric Multispectral Imaging (HMI) technique was also applied in selected areas of two panels. Due to the accurate calibration system, this technique is able to obtain high-precision colorimetric and reflectance measurements, which can be repeated for proper surface monitoring. The integrated analysis of the different wavelengths' images—in particular, the ones processed in false colors—made it possible to distinguish the portions affected by retouching or repainting and to recover the legibility of some figures that showed chromatic alterations of the original pictorial layers. The IR reflectography, in addition to highlighting the portions that lost materials and were subject to non-original interventions, emphasized the presence of the underdrawing, which was detected using the *spolvero* technique. UVF photography led to a preliminary mapping of the organic and inorganic materials that exhibited characteristic induced fluorescence, such as a binder in correspondence with the original azurite painting or the wide use of white zinc in the retouched areas. The collected data made it possible to form a better iconographic interpretation. Moreover, it also enabled us to accurately select the areas to be investigated using spectroscopic analyses, both *in situ* and on micro-samples, in order to deepen our knowledge of the techniques used by the artist to create the original painting, and to detect subsequent interventions.

**Keywords:** UV fluorescence (UVF); IR reflectography (IRR); IR false colors (IRFC); hypercolorimetric multispectral imaging (HMI)

## 1. Introduction

This paper contributes to the special issue, "Results of the II National Research project of AIAr: archaeometric study of the frescoes by Saturnino Gatti and workshop at the church of San Panfilo in Tornimparte (AQ, Italy)". This special issue collates the scientific results of the II National Research Project, which was conducted by members of the Italian Association of Archaeometry (AIAr). For in-depth details on the aims of the project, see the introduction of the special issue [1].

The imaging analyses conducted on the wall paintings attributed to the Italian master Saturnino Gatti in the apse of the Church of San Panfilo (Tornimparte—AQ, Italy) were executed with two main purposes. First, after visualizing and identifying the areas affected by surface degradation, we aimed to map the different colored/pigmented areas in order

to expand upon the amount of information available with regard to the materials used; this information was obtained using spectroscopic techniques [2–4]. The imaging techniques also allow us to obtain information concerning the primary technique used on the wall paintings, which provides art historians and conservators with useful information for historical reconstructions of the different phases of the artwork [5–8]. Imaging analysis has the advantage of making information accessible in a simple way, via the rendering of images and maps, which, through the use of colors, chromatic gradients, and the spatial distribution of values, allows for an immediate understanding of the data. For this reason, techniques that use exciting radiation, from near ultraviolet (365 nm) to near infrared (1000 nm), have been employed, thus enabling images to be acquired in different bands of the visible and near infrared range (400–1100 nm) [9–11].

Ultraviolet radiation can excite certain pigments and materials used during artistic production, causing them to emit light in the visible range, based on their specific composition. This phenomenon of luminescence can reveal important details about the artwork that may be barely visible, not visible at all, or visible under normal lighting conditions. Indeed, Ultraviolet Fluorescence (UVF) photography is a very useful documentation technique in the field of cultural heritage, especially when it occurs before any restoration work has taken place. This is because it allows us to obtain relevant information concerning the level of conservation that has taken place on the surfaces of the paintings, and it allows us to ascertain which materials were used; indeed, it can differentiate between original and added materials, such as those used during grouting and retouching processes. UVF photography is also useful for distinguishing between classes of materials that appear similar in visible light, but different under UV radiation. Moreover, it can also detect traces of materials that are no longer observable with the naked eye [12–15]. For these reasons, UVF photography is widely used in the field of restoration, and in general, it is used for the evaluation of the general state of conservation, with regard to artworks.

Infrared reflectography (IRR), as with UVF photography, is a non-invasive imaging analytical technique used for analysis. It is widely used in the conservation field and in the restoration of art objects [16]. This technique, based on the transparency of the different pigments when placed under infrared radiation, can reveal hidden features underneath the pictorial layer, such as underdrawings, *pentimenti*, and changes made by the artist during the painting process. The information obtained through IRR can be crucial for restorers, and it can help them make decisions concerning the conservation, restoration, or treatment of a painting; for example, infrared radiation can show possible material differences between pigments that visually appear similar, or it can highlight the stratified use of pigments with different opacities. Infrared reflectography can also provide valuable insights into the artist's working methods and creative process, reveal details of the artist's primary techniques via the underlying drawings, as well show *pentimenti* and changes in the settings of the scenes; indeed, it is possible that this technique can reveal the presence of previous versions of the painting [13,17].

By combining images acquired using infrared and visible light, it is possible to merge information deriving from different regions of the electromagnetic spectrum; thus, images are obtained in false colors which allow us to better discriminate between characteristics of the painting that would appear less accentuated compared to the response of the IRR image alone. When applied to wall paintings, infrared false color (IRFC) imaging can help with the identification of areas affected by damages such as detachments, cracking, discoloration, or delamination of the surface layers. Similarly to IRR, it can also help to identify the presence of underlying layers of paint or other materials that may have been covered by subsequent overpaintings or modifications; it has the advantage of presenting areas that appear irregular in different colors, rather than in grayscale, such is the case with reflectography [14,18].

Finally, hypercolorimetric multispectral imaging (HMI) was tested on two areas of the paintings. This multispectral imaging technique, developed and patented by the Roman society, Profilocolore, has been widely used to investigate easel paintings, but it has rarely

been used on wall paintings; for this reason, taking the advantage of Tornimparte research project, HMI was applied in order to acquire two portions of the wall paintings and to post-process the obtained images. HMI is a powerful multispectral technique that, due to the image calibration procedure, allows us to obtain final images with a high level of reflectance and colorimetric precision [19]. The output after calibration consists of monochromatic images, to which several available algorithms in the processing software can be applied. These algorithms relate to the following: reflectance and chromatic comparison and mapping; normalized differences between monochromatic bands; principal component analysis; and the production of false color images (both infrared and ultraviolet) by simply combining the RGB channels with one of the IR bands or the UV bands [20].

Overall, the interpreted results concerning the different imaging techniques provided synergic information that was useful for verifying and mapping the conservative state of the painted surfaces, as well as obtaining preliminary information on the materials used. Such information will be expanded through the application of spectroscopic techniques performed directly *in situ*, in particular, X-ray fluorescence (XRF), Fourier Transform Infrared Spectroscopy (FTIR), Raman Spectroscopy, Fiber Optics Reflectance Spectroscopy in the UV–Vis–NIR Range (FORS) [21], and in laboratory analyses performed on samples [22].

## 2. Materials and Methods

Each imaging technique helps document and reveal specific and precious data, thus leading to an increased understanding of the painting's layout and history.

Image analyses were performed on 3 of the 5 panels of the lower part of the apse that still had painted layers (Panel A—the traitorous kiss of Judas and the capture of Christ in the Garden of Olives, Panel D—Deposition of Christ and Panel E—The Resurrection), as well as on the vault where the scene of the glory of God is represented. A detailed description of the painted scenes in the apse of St. Panfilo church is reported in the literature [23,24] and in a recently published paper [25]. The HMI analyses were instead focused on the flat areas of panels A and E. For a more extensive discussion of the site and of the project, refer to Galli et al. [1]. All the images were acquired from the ground with the aid of a tripod, exploiting different angles to obtain the details of the paintings; the light sources were positioned from time to time, as needed, supported by stands.

### 2.1. Diffuse Visible Light Photography (Vis)

Photography in diffuse visible light is always useful and often necessary when carrying out analyses of images; this allows both the documentation of the actual state of the surfaces at the time the analyses were executed, and a continuous comparison during the processing and post-production phases of the images that were acquired using other techniques. Diffuse visible light photography was conducted with an Olympus camera that had a 16 Mpx sensor and a 40–150 mm lens. Two halogen lamps were used as light sources.

### 2.2. Ultraviolet Fluorescence Photography (UVF)

For the Ultraviolet Florescence Photography, the images were obtained using a Nikon D800 digital reflex camera modified in Full Range to let the sensor grow accustomed to the electromagnetic spectrum from approximately 300 to 1000 nm. In front of the lens, two filters were applied: the filter A and the UV-IR cut filter, the spectra of which are shown in Figure 1.

**Spectrum PFCL A**

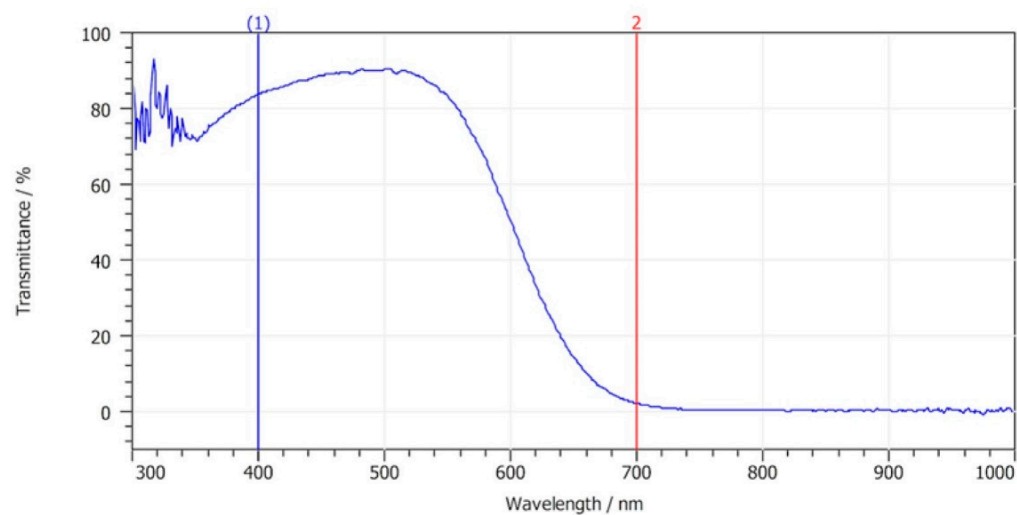

**Spectrum PFCL B**

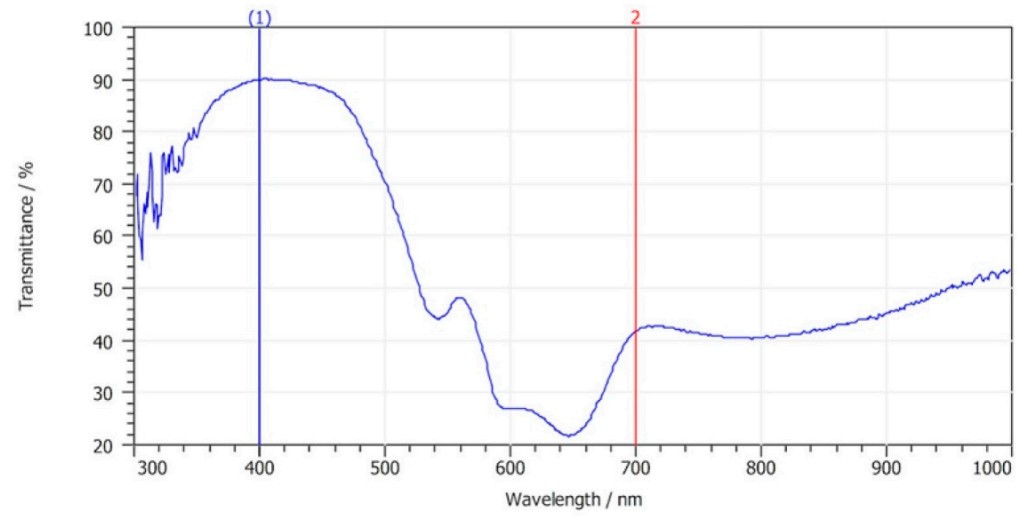

**Spectrum PFCL UV IR cut**

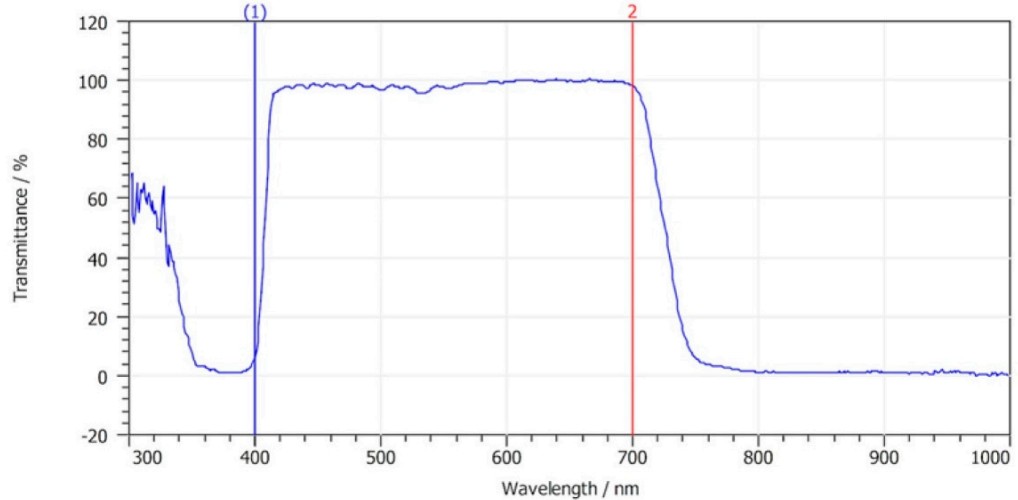

**Figure 1.** Spectra of cut filters, A, B, and UV-IR, used in HMI and UVF.

The UV radiation was obtained using four CR230B-HP 10W LED projectors, at peak emission at 365 nm, which were mounted at 45° in relation to the camera. LED projectors allowed us to avoid the blue-violet background that is generally obtained when using traditional UV lamps or tubes. The other advantage to using these LED projectors is that they are powered by an internal rechargeable battery; especially in the case of large surfaces, as with the case of the St. Panfilo church, they can be moved easily without the necessity of paying attention to the power supply cables or any extension cables [26].

### 2.3. Infrared Reflectography (IRR)

The reflectographic infrared images were acquired using a Sony H50 digital camera with a 9 Mpx sensor and integrated optics, with an equivalent focal length of 31–465 mm. The camera was equipped with a switch that allowed us to manually remove the UV-IR cut filter placed in front of the sensor to obtain a full range in terms of sensitivity, without having to disassemble the camera. To record infrared images, an 850 nm high-pass filter was placed on the lens to limit the sensitivity range of the sensor between 850 and 1100 nm. The infrared radiation was produced by two 1000 W halogen lamps positioned at about 45° in relation to the surface in order to make the lighting homogeneous and to minimize reflections [27].

### 2.4. Infrared False Color (IRFC)

The Infrared False Color images were processed in two different ways: using Hypercolorimetric Multispectral Imaging (see Section 2.5) and by combining the visible and infrared images acquired with the Sony H50 camera.

After manually removing the UV-IR cut filter on the sensor of the Sony H50 camera, it was possible to acquire two (almost completely identical) pictures, with the same level of detail, in visible and infrared light. An image processing program was used to perfectly align the visible and infrared images, using some details inside the images themselves as references. Then, the channels were recombined so as to obtain the false color output image. The procedure was structured as follows: starting with the digital image, it was initially necessary to break it down into the three main channels (red, green, and blue); then, in order to replace the data contained in the channels, that which was recorded in the infrared range was inputted into the red channel, the information previously contained in the red channel was inputted into the green channel, and the information previously contained in the green channel was inputted into the blue channel. Thus, we moved from an image composed of red, green, and blue, to one composed of infrared, red, and green, by shifting the information within the three channels. [28]

### 2.5. Hypercolorimetric Multispectral Imaging

HMI acquisition was performed only on the flat surfaces, in accordance with a procedure described in previously published papers, and here, it was briefly summarized [29–31]. It used the same Nikon D800 digital reflex camera, modified in Full Range, during the UVF analyses.

1.  The first step involves the acquisition of two images; one was acquired using filter A and the other using filter B (spectra shown in the Figure 1). The filters were screwed in front of the camera lens before each shot was taken. The lighting was set up using NEEWER (Neewer, Shenzhen, China) 750II Flashes Speedlite TTL with an LCD Display and Wireless Triggers. The flashes were modified by removing their front plastic lenses, thus allowing emissions in the 300–1000 nm region. To produce radiometrically and colorimetric calibrated images, white patches and a color-checker (36 colour samples from the Natural Color System® ©, NCS catalog) were positioned in the scene around the painting.
2.  The second step concerns the calibration of the two acquired images using the proprietary software, SpectraPick® (v1.1, created by Profilocolore, Rome, Italy), that, at the end of the process, produces seven tiff files representing the multispectral

monochromatic images centered at 350 nm (UVR), 450 nm, 550 nm, 650 nm, 750 nm (IR1), 850 nm (IR2), and 960 nm (IR3), as well as the RGB 16-bit color image [32,33].

The third step of the HMI system involves the processing of the calibrated images using the software, PickViewer® (v1.0, created by Profilocolore), which provides several tools with which to obtain infrared and ultraviolet false color images, to read pixelwise colorimetry and spectral reflectance, to create similarity maps according to color or spectral data, to apply principal component analysis (PCA), and so on. The results can be saved as an image in tiff, png, or jpeg format.

### 2.6. Optical Microscopy in Visible Light (OM)

To document the restricted areas to be analyzed using portable spectroscopic techniques, a digital microscope with a 5 Mpx CCD sensor was also used in visible light. Using the microscope software calibration tool in conjunction with the reference target, it was possible to define a metric reference for each magnification level.

## 3. Results and Discussion

### 3.1. Material Mapping, Retouching, and Remaking

The UV fluorescence photography led to a preliminary mapping of the surfaces, which differentiated between the use of organic and inorganic materials; these presented with a characteristic fluorescence. First, it was possible to identify the presence of a significant quantity of zinc white, a pigment in use since the late eighteenth century, and therefore, associated with areas affected by pictorial retouching [33]. Indeed, the presence of zinc (Zn) was also detected by X-Ray fluorescence (XRF) in the punctual analyses of the restored areas [2]. Zinc white pigment is clearly visible due to its lemon-yellow fluorescence in the lower part of panel A, representing the betrayal of Judas. It is also visible under the window, which appears as a bright blue color in the image due to a paper which served to shield the apse from the outdoor sunlight (Figure 2) [34]. In the upper part of the same panel, especially in the background, a light yellow fluorescence is visible; this could be associated with the azurite binder, a pigment which has been recognized in the blue areas via spectroscopic analyses with XRF due to the clear presence of copper (Cu) [2]. This same fluorescence can also be observed in panel 4, the last from the left side of the apse conch, where the resurrection of Christ is represented (see Figure 3A). Areas that appear white in visible light show a faint light blue fluorescence (Figure 3B); this may be associated with calcium carbonate white, contrary to the hypothesis of the art historians that assumed that it was the use of lead white, used in the highlights of the paintings.

A detail was acquired in panel 4 wherein some fluorescent lines were observed in the general view of the wall (Figure 3A). The image in Figure 3B shows how the rays that emanate from the angels are characterized by a yellow-orange fluorescence which could be due to the presence of organic material. In accordance with art historians, the presence of possible original gilding work was hypothesized, and therefore, the organic substance whose fluorescence is observed could be referred to as the so-called *missione*, which was used to make the adhesive for the gold. Optical microscopy revealed the presence of golden traces in the rays (Figure 3C); indeed, the X Ray Fluorescence analyses, performed after the first round of image analyses, confirmed the theory that gilding had taken place. In fact, XRF spectroscopy detected the presence of gold (Au) as a chemical element that characterized those same areas [2]. The glue used for gilding was generally made of siccative oil and terpene resin; therefore, the yellow fluorescence may be associated with the lipidic component of the *missione* [35–37].

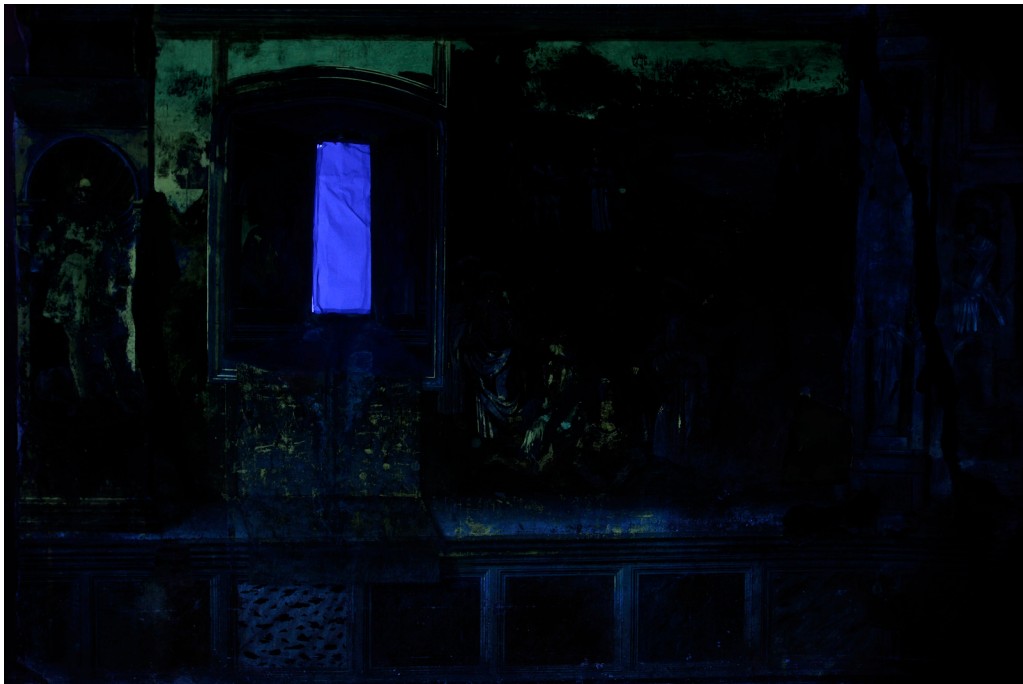

**Figure 2.** The UVF image of panel A, Garden of Olives.

Infrared Reflectography, together with the Infrared False Colors, allowed us to distinguish between the portions of the painting that were affected by pictorial restorations. Pigments appearing similar in visible light may exhibit a different response to infrared radiation, which is rendered at a different level of brightness in reflectographic images; this difference results in a different color response in the false color infrared images. An example is shown in panel A, in Figure 4B, in which the restoration areas in the green land are identifiable as spots with lighter levels of gray, compared with the surrounding areas. This restoration intervention is also confirmed in false color images (Figure 4C), in which the same spots exhibit a pinkish color that may be attributable to the original materials, unlike those exhibiting a green color.

Once again, via the false color images, it was possible to recognize small portions of paintings in which different materials were used. Compared with most areas, which appeared chromatically similar to the naked eye, as in the case of the blue near the Virgin's head in the scene of the deposition (Figure 5C), a single fragment shows a pink-red response in the false color image. These results are also relevant because they guided the decision concerning which points should be investigated using spectroscopic techniques [2].

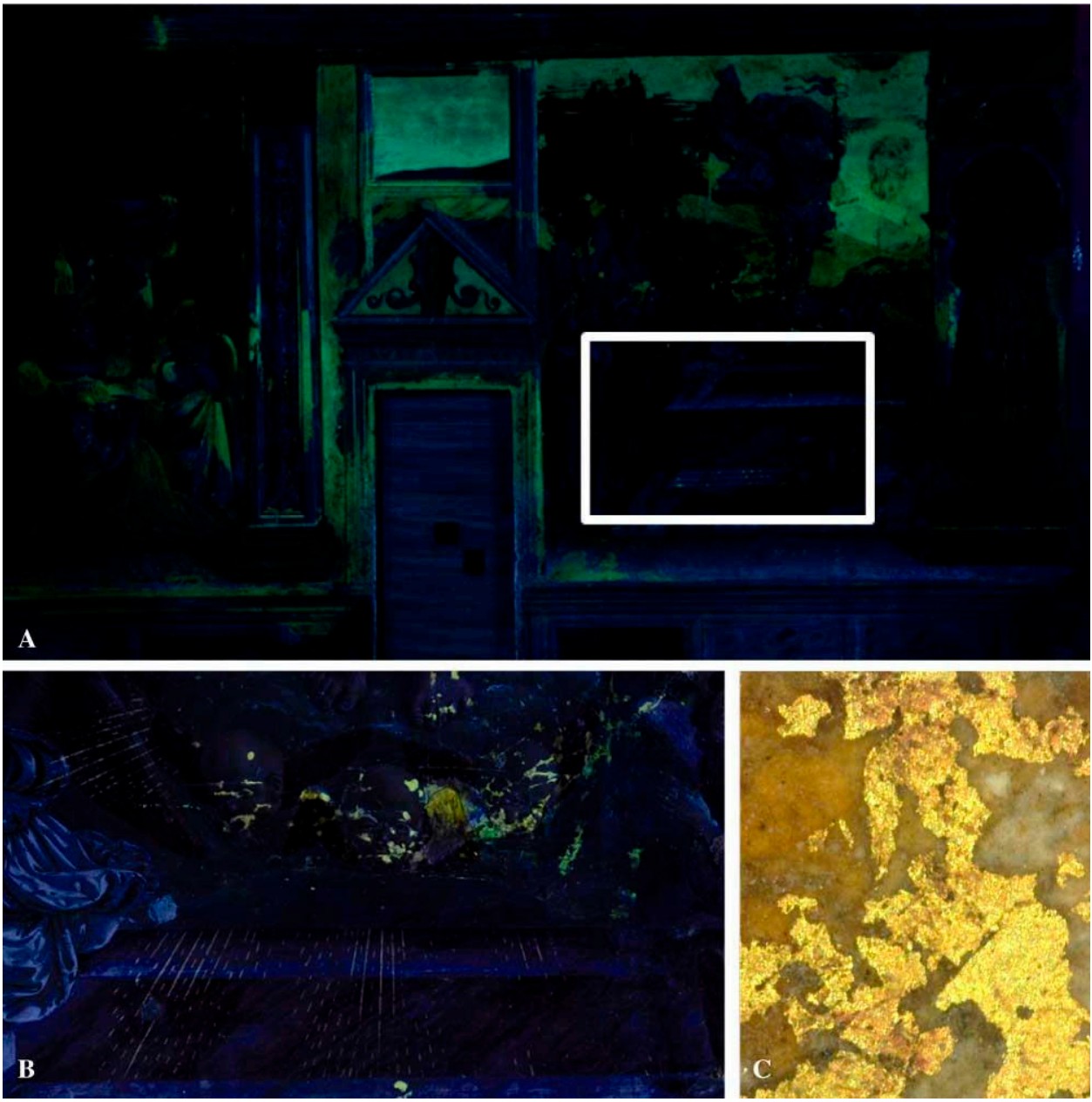

**Figure 3.** The UVF image of panel E, the Resurrection: (**A**) general view; the white rectangle indicates the detail shown in (**B**). (**C**) Detail of the golden traces found via OM in the rays.

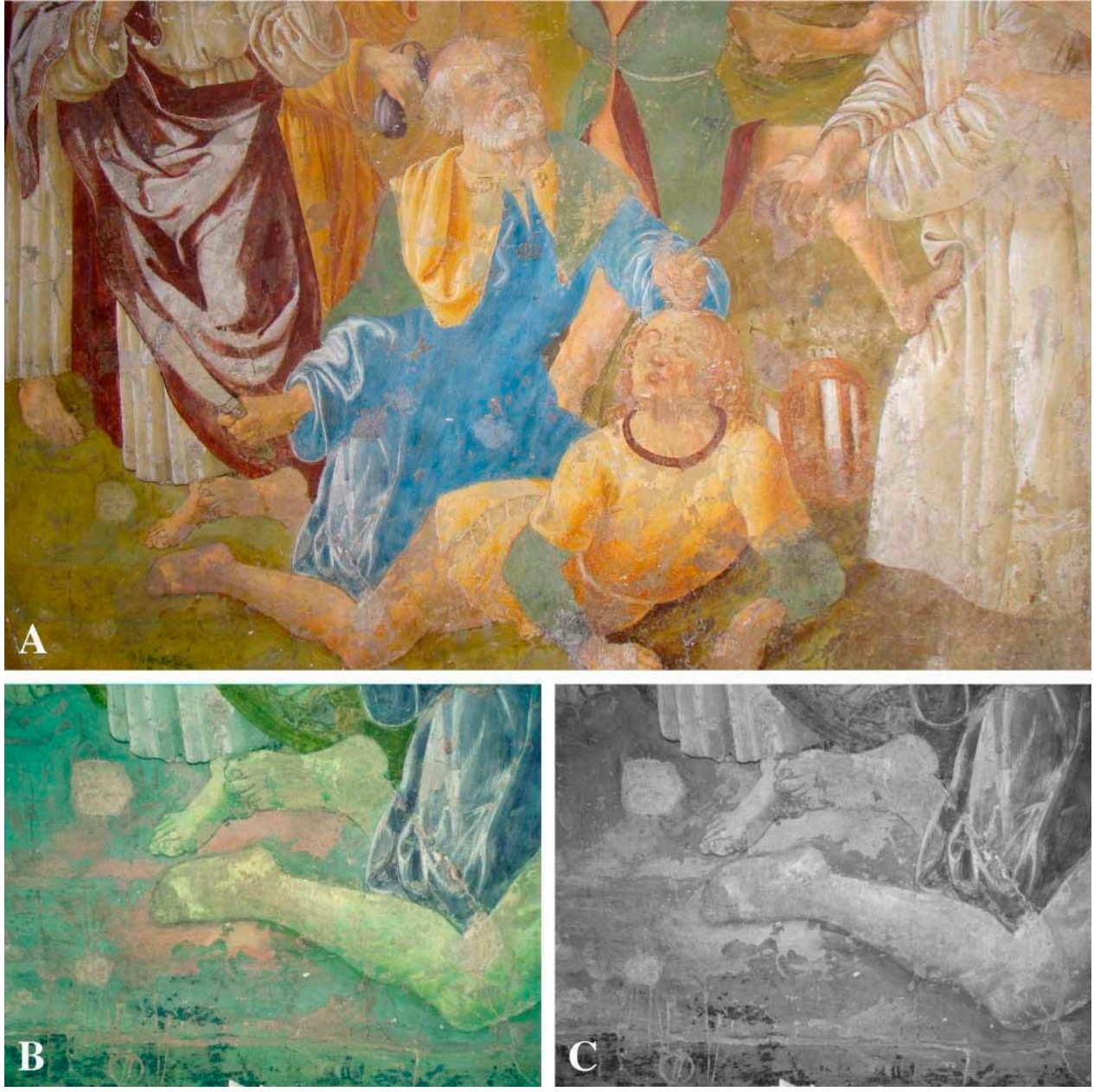

**Figure 4.** Detail of Panel A, Garden of Olives (**A**) in visible light, (**B**) IRR, and (**C**) IRFC.

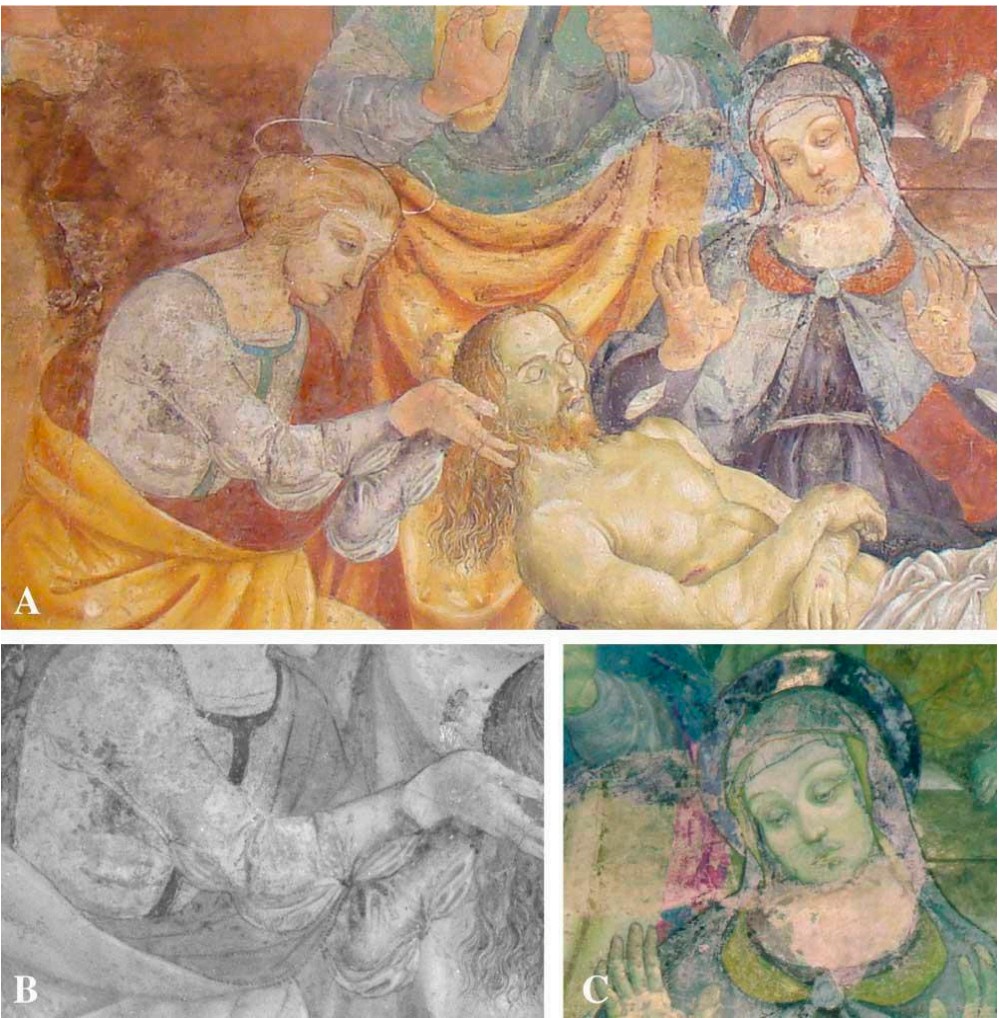

**Figure 5.** Detail of panel D, Deposition of Christ (**A**) in visible light, (**B**) IRR, and (**C**) IRFC.

### 3.2. Underdrawing

IR reflectography, in addition to allowing the mapping of the surface portions affected by material losses and highlighting the presence of restoration material, has increased the amount of information relating to artistic techniques; in particular, it proffers insight into the methods used by the artist to draw the different figures represented in the pictorial cycle [38]. Along the perimeters of the various characters, dots precisely defining the contours and the main constructive features are visible; it is known that reflectography allows us to identify underlying drawings made with materials that are opaque when examined with infrared radiation, and they are generally carbon-based [39]. The presence of the dots may therefore be due to the transfer of the drawing from cartoons or sheets via the "*spolvero*" (punching) technique. This last technique involves the creation of a full-sized drawing on a preparatory cartoon; then, the contours of the different figures are subsequently perforated with a needle or another small tip. Next, the perforated cardboard is placed on the wall's surface, and the perforated parts are dabbed with a canvas bag filled with ash, charcoal, or other fine dust that may leave a trace on the wall [40]. A clear example of such dusting is visible in image 5B, in which the series of dark dots are clearly defined near the forearm and the folds of the dress of the woman identified as Mary Magdalene [24], who supports the head of the dead Christ. Similarly, the signs of the transfer are visible on her cloak. Evidence of the same type of drawing method has been found in the other panels of the apse; this acts as evidence of a methodological unity in the realization of the entire pictorial cycle, and in support of the attribution of the artwork itself. Once again, when carefully comparing Figure 5A (visible image) with Figure 5B

(reflectography), it can be seen that in some parts, such as along the shoulders, a brown pigment was used to enhance the detail of the contours, thus preventing the possibility of detecting the underlying drawing via the IRR technique.

### 3.3. HMI Results

To test the potential of HMI techniques, the acquired images were focused on two detailed areas of the panels A and E, as they were the only panels with flat surfaces.

The software PickViewer®, included in the HMI system, allowed us to obtain the infrared false color (IRFC) output by simply combining the RGB channels with one of the three IR channels. An example is shown in Figure 6. The retouched lacunae exhibit a pink fluorescence that better distinguishes them from the original parts. The blue dress has a dark blue response in IRFC, thus suggesting the possible presence of azurite (hypothesis confirmed via punctual analyses) [2]. In this same area, other processes were performed using the PickViewer® tools. The chromatic similarity algorithm was applied on the blue dress, as shown in Figure 7). This tool compares the values of the chromatic coordinates of the selected point (on the RGB image) with all pixels in the image, and it produces a B/W image where the white pixels have the same chromatic values as the chosen point, and the black pixels are completely different in terms of color data.

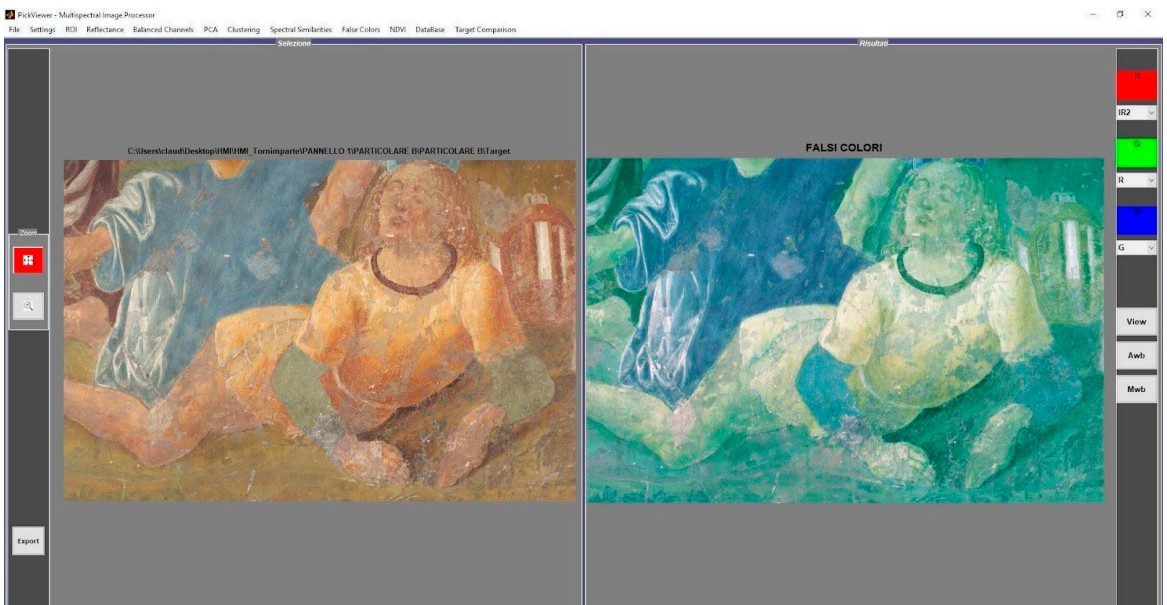

**Figure 6.** The graphic user interface (GUI) of the PickViewer®, shown on the **left** of the RGB image, and on the **right**, the IRFC result is shown. Detail of panel A.

In Figure 7, the black areas of the dress identify the lacunae of the painting layer or the highlights used to outline the drapery of the dress.

In the chosen area of panel A, further processes were performed, in particular, principal component analysis (PCA) was applied using the three infrared bands (IR1, IR2, and IR3). The result is shown in Figure 8, wherein the first PC highlights the drawing in the feet, in the hand, and above the sword hilt and blade (Figure 8).

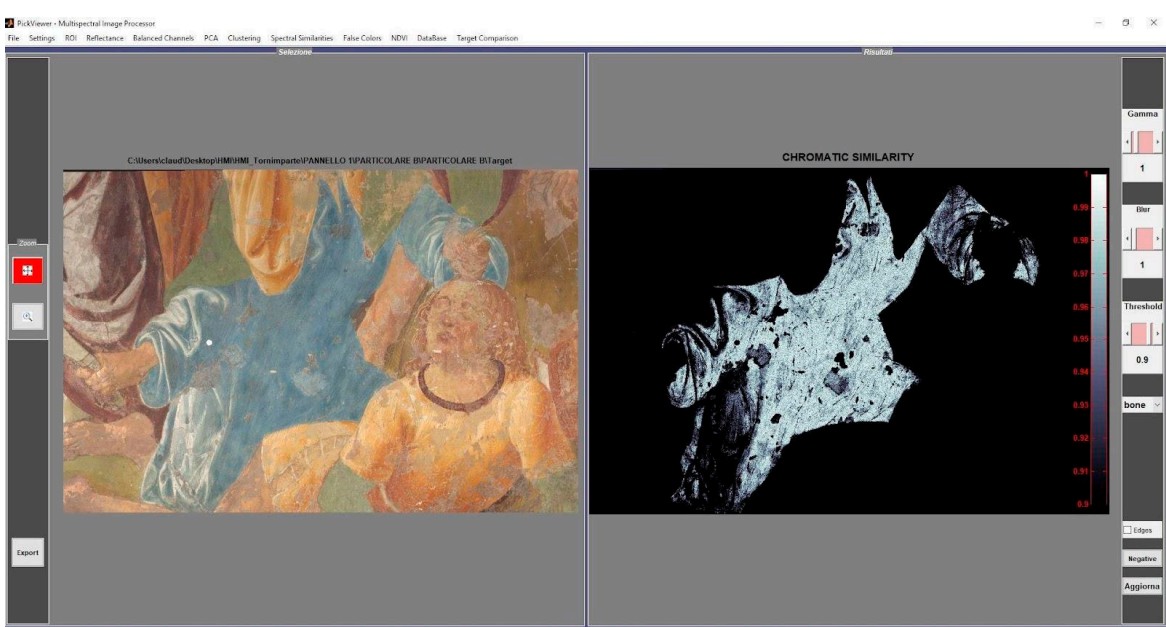

**Figure 7.** The GUI of the PickViewer® shows the RGB image on the **left**, with the selected point (white dot) for the application of the chromatic similarity tool, and on the **right**, the result is shown. Detail of panel A.

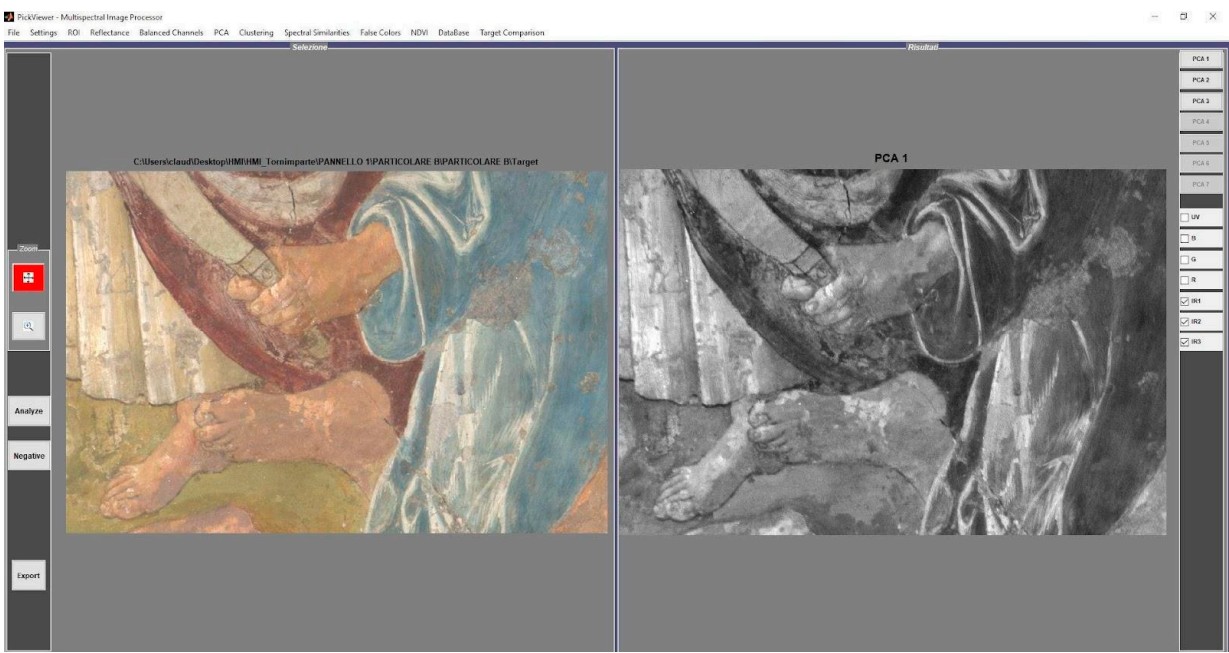

**Figure 8.** The GUI of the PickViewer® shows the RGB image on the **left**, and the PC1 obtained by applying the PCA to the three IR bands is shown on the **right**. Detail of panel A.

The chromatic similarity tool was also applied to a green point in panel E (Figure 9). This green, in the leg of the character that is lying down, seems to be the result of a retouch. The result of the application of the tool shows that the green retouches are located in the legs, but also in the upper part of the selected area, thus corresponding with the background. The chromatic similarity tool was also applied to a green area that seems to be the original (Figure 10).

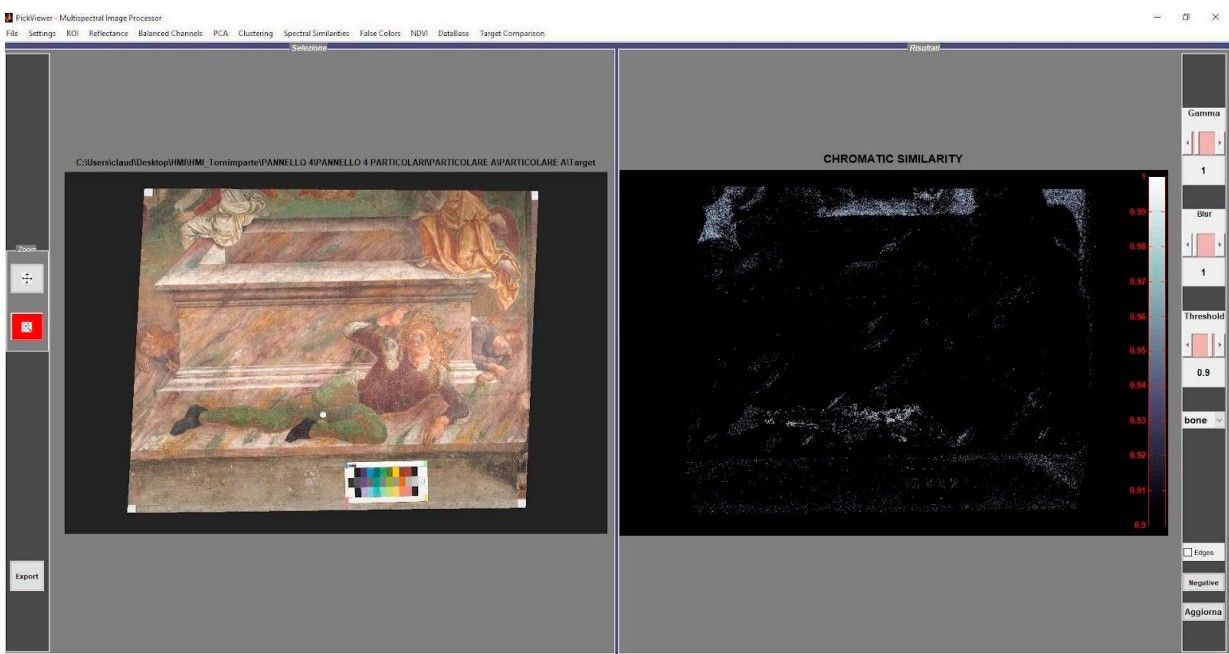

**Figure 9.** The GUI of the PickViewer® shows the RGB image with the selected point (white dot on the leg of the character that is lying down, probably non-original panting) for the application of the chromatic similarity tool on the **left**, and the result is shown on the **right**. Detail of panel E.

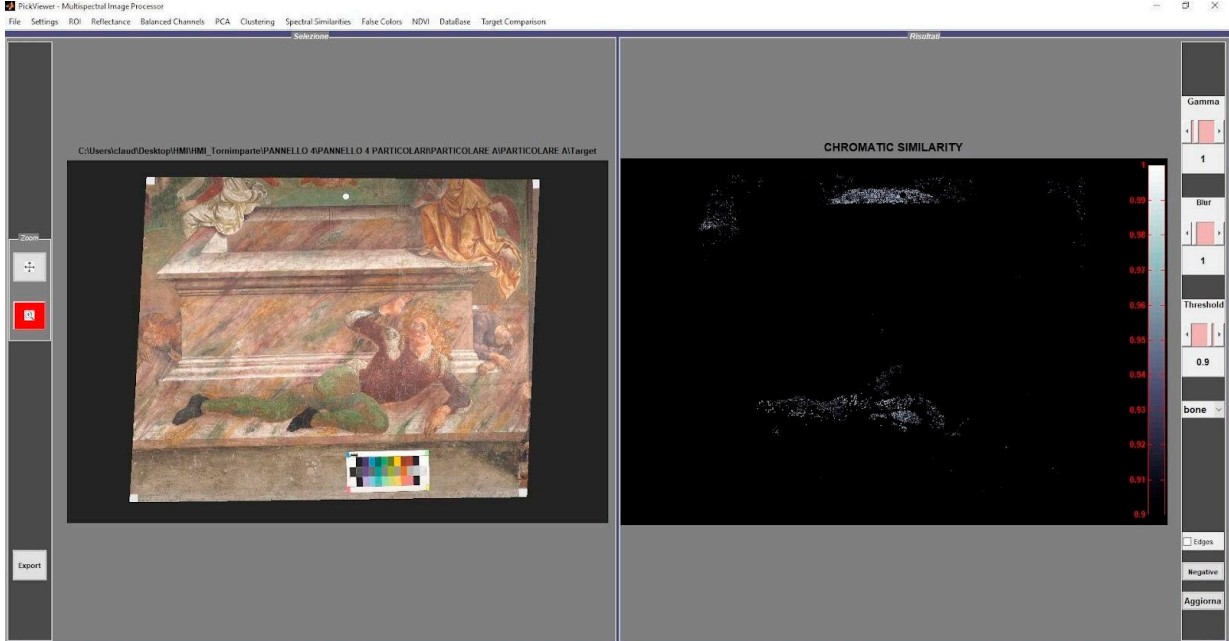

**Figure 10.** The GUI of the PickViewer® shows the RGB image with the selected point (white dot in the upper part of the green area, probably original) for the application of the chromatic similarity tool on the **left**, and the result is shown on the **right**. Detail of panel E.

The result of the tool's application highlights that the green in the upper part of the selected area, which corresponds with the top of the sepulcher, has a chromatic similarity with that of the legs of the lying gendarme and a small area of land on the left side, but not with the green areas that are presumed to be non-original (mapped in the Figure 9).

## 4. Conclusions

The data acquired in this phase of the Tornimparte project led to various results in terms of awareness, regarding the data available to progress the activities, the transition to the next steps of the investigation, and the materials and techniques used to create the wall paintings by Saturnino Gatti. First, the mapping of different pictorial materials made it possible to proceed, with greater awareness, with the decision concerning which points should be used for non-invasive spectroscopic analyses that can characterize the materials of the wall painting at the atomic and molecular level. Even the choice of micro-sampling the painting layers, which were used in the laboratory analysis, was guided by the acquired multispectral images. This approach made it possible to distinguish between the areas wherein the materials could be attributable to the original choices of the artist, and those that were subject to overpainting or restoration interventions. The use of a multispectral approach, with radiation ranging from ultraviolet to near infrared, highlighted the presence of specific materials, such as zinc white, and the recognition of several different materials in areas that appeared chromatically homogeneous.

The imaging techniques also allowed us to confirm some hypotheses concerning the executive techniques of Saturnino Gatti; the gilding, for example, was confirmed in several areas wherein the presence of visible fluorescence induced by UV radiation can be associated with the organic material used to glue the gold leaf. Small traces of the same gold leaf have been detected via digital optical microscopy at different points, such as in the rays in the risen Christ or in the halos of the saints.

In addition to the information regarding the materials of the artwork, it was also possible to provide a precise mapping of the level of conservation that had been conducted on the surfaces; it was possible to identify the presence of restoration materials, as well as the various degradation phenomena that were taking place.

Finally, as well as answering the initial research questions, it was also possible to provide information on the primary technique used to create the wall paintings; in particular, the method to transpose the drawings of the various figures to the wall. Indeed, the presence of the "spolvero" (punching) technique was unknown at the beginning of the investigation, and it provided important information on how the iconographic and scenic system was executed; in this case, a transfer technique based on the use of paper and/or cardboard was used.

**Author Contributions:** Investigation, L.B., S.C., A.G., C.P. and L.L.; Data Curation, L.B., S.C., A.G., C.P. and L.L.; Writing—Original Draft Preparation S.C. and C.P.; Writing—Review and Editing, L.B., S.C., A.G., C.P. and L.L.; Supervision, A.G. All authors have read and agreed to the published version of the manuscript.

**Funding:** This work was performed as part of Project Tornimparte—"Archeometric investigation of the pictorial cycle of Saturnino Gatti in Tornimparte (AQ, Italy)" sponsored in 2021 by the Italian Association of Archeometry AIAR (www.associazioneaiar.com).

**Data Availability Statement:** The data presented in this study are available on request from the Italian Association of Archeometry AIAR (www.associazioneaiar.com).

**Acknowledgments:** The authors want to thank the Proloco Tornimparte association and the mayor of the municipality of Tornimparte for their availability and support shown during the diagnostic campaign.

**Conflicts of Interest:** The authors declare no conflict of interest.

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
