# Peer review of "Materials and Technique: The First Look at Saturnino Gatti"

_applsci, doi:10.3390/app13116842_

Round 1

Reviewer 1 Report

The paper is well written and structured in the presentation of the results. Some puctual analyses are mentioned in the text but no always specified. Please ammend this at page 3 (where punctual spectroscopic techniques are mentioned) or page 10 (again, what kind of punctual analyses, see first line).

Author Response

Thanks for the appreciation for our work, we have specified the spectroscopic analyses with which the surfaces were investigated the first time we mention them, on page 3.

Reviewer 2 Report

The manuscript titled: “Materials and technique: the first look at Saturnino Gatti” presents an interesting study of the pictorial cycle by Saturnino Gatti, focusing on the painting materials and pigments used. The authors are using image analyses and spectroscopic techniques in order to examine the general condition of the conservation of surfaces, as well as to identify and map the use of the different pigments and painting materials. Through the collected data by the analysis methods and the techniques used, the authors made an iconographic interpretation and selected specific areas for more detailed spectroscopic analysis both in situ and on sampling. This detailed study allowed them to identify more accurately and in-depth the techniques used for the realization of the original painting as well as the subsequent reconstructions. The manuscript corresponds to the aims of the Journal “Applied Sciences”. The objective is clear and accomplished as there are no doubtful or controversial arguments. The key findings and conclusions of the research are supported by the presented results. The manuscript is well-structured and has the appropriate length. Figures and Tables are well-displayed and sufficiently described. There are some differences in the font style of the figure labels, that need to be corrected to be uniform. In some sections the appropriate citations have been used, although there seems to be inadequate bibliography in other sections of the paper. English needs some improvements, as some grammar and syntax errors are detected, while some phrases and sentences are informal for a research paper. Since there are still a few corrections that should be made prior to publication in “Applied Sciences”, Minor Revision is suggested.

Significant comments:

General comment (optional): Please consider altering the title of the manuscript, for example from “Materials and technique: the first look at Saturnino Gatti” to “The first look at Saturnino Gatti: Materials and Techniques”. It is preferable to first mention the subject of research, and then the materials and techniques (plural) used.

General comment: Please review your English. There are some grammar and syntax errors detected in the manuscript, and some sentences or phrases are informal for a research paper.

Figure labels: Please use a uniform font type for the figure labels throughout the manuscript. The labels of Figure 2-7 are bold, while of Figure 1, 8-10 are not bold.

Figure 1: Please use a bigger font in the axis labels and numbering of the presented spectra, or increase the size of the images in order to be more clearly readable.

General comment: There seems to be inadequate bibliography in many sections of the paper. Please consider including more citations where needed.

Comment for the English language:

Please review your English. There are some grammar and syntax errors detected in the manuscript, and some sentences or phrases are informal for a research paper.

Author Response

Thank you for your appreciation for our work, we have taken into account the notes indicated.

General comment (optional): Please consider altering the title of the manuscript, for example from “Materials and technique: the first look at Saturnino Gatti” to “The first look at Saturnino Gatti: Materials and Techniques”. It is preferable to first mention the subject of research, and then the materials and techniques (plural) used.

Reply
This work is part of a special issue entirely dedicated to the study of Saturnino Gatti's wall paintings. The title has already been agreed with the editors of the special issue and cannot be changed. To clarify, “Materials and technique” refer to those of the artist and not to the methods of investigation employed by the authors of the paper.

General comment: Please review your English. There are some grammar and syntax errors detected in the manuscript, and some sentences or phrases are informal for a research paper.

Reply
We have corrected and revised some sentences

Figure labels: Please use a uniform font type for the figure labels throughout the manuscript. The labels of Figure 2-7 are bold, while of Figure 1, 8-10 are not bold.

Reply
We have corrected the labels by using bold for the figures.

Figure 1: Please use a bigger font in the axis labels and numbering of the presented spectra, or increase the size of the images in order to be more clearly readable.

Reply
We have enlarged the images for a better readability.

General comment: There seems to be inadequate bibliography in many sections of the paper. Please consider including more citations where needed.

Reply
We have provided to insert new bibliography where necessary.